# BOOSTING REINFORCEMENT LEARNING IN 3D VISUOSPATIAL TASKS THROUGH HUMAN-INFORMED CURRICULUM DESIGN

## ABSTRACT

Reinforcement Learning is a mature technology, often suggested as a potential route towards Artificial General Intelligence, with the ambitious goal of replicating the wide range of abilities found in natural and artificial intelligence, including the complexities of human cognition. While RL had shown successes in relatively constrained environments, such as the classic Atari games and specific continuous control problems, recent years have seen efforts to expand its applicability. This work investigates the potential of RL in demonstrating intelligent behaviour and its progress in addressing more complex and less structured problem domains.

We present an investigation into the capacity of modern RL frameworks in addressing a seemingly straightforward 3D Same-Different visuospatial task. While initial applications of state-of-the-art methods, including PPO, behavioural cloning and imitation learning, revealed challenges in directly learning optimal strategies, the successful implementation of curriculum learning offers a promising avenue. Effective learning was achieved by strategically designing the lesson plan based on the findings of a real-world human experiment.

## 1 INTRODUCTION

Reinforcement learning (RL) has its roots in behavioral psychology, notably Edward Thorndike's Law of Effect Thorndike (1898; 2017), which states that actions followed by satisfying outcomes are likely to be repeated. B.F. Skinner formalized this into Operant Conditioning Skinner (1938; 2019), introducing reinforcement schedules to systematically shape behavior. Foundational algorithms such as Q-learning and Policy Gradients Watkins & Dayan (1992); Sutton et al. (1998) propelled the field, and its integration with deep learning led to major breakthroughs. These include achieving superhuman performance in complex games like Atari, Go, and StarCraft II Mnih et al. (2013); Silver et al. (2017); Vinyals et al. (2019), solving scientific problems like protein folding with AlphaFold Varadi et al. (2022), and aligning language models via RLHF Christiano et al. (2017).

This progress has fueled speculation that reward-driven learning is a potential path toward artificial general intelligence (AGI): *"Reward is enough to drive behaviour that exhibits abilities studied in natural and artificial intelligence, including knowledge, learning, perception, social intelligence, language, generalisation and imitation. (...) reinforcement learning agents could constitute a solution to artificial general intelligence."* Silver et al. (2021).

We investigate this potential by applying RL to the Same-Different task—a primitive psychophysical skill fundamental to visuocognition Hautus et al. (2021). While traditionally studied with 2D stimuli, we focus on a 3D version by replicating a recent real-world experiment Solbach & Tsotsos (2023) in a Unity simulation. We evaluate several RL approaches for this task, including PPO Schulman et al. (2017), GAIL, Behavioral Cloning Bain & Sammut (1995), and Curriculum Learning Bengio et al. (2009).

## 2 PROBING VISUOSPATIAL PERCEPTION

The Same-Different task, where an observer must determine if two stimuli are identical, is considered one of the most primitive human psychophysical tasks and serves as a building block for more

complex cognitive operations Hautus et al. (2021). Our choice of this task is motivated by its historical prominence in cognitive science, particularly the influential study by Shepard and Metzler on mental rotation Shepard & Metzler (1971). Using 2D images of 3D shapes (Figure 1), they discovered a linear correlation between an object's rotational difference and the time subjects required to confirm its identity – a foundational principle in the study of visuospatial reasoning (Bricolo et al. (1996); Biederman (2000)).

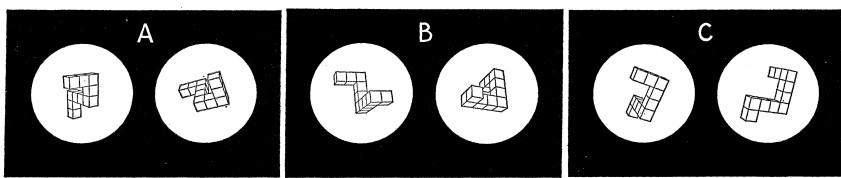

Figure 1: The stimulus used by Shepard & Metzler (1971) to assess the human cogntive ability of mental rotation of three-dimensional objects. Note how the stimulus is an image and depicts a 3D object.

However, a key limitation of classic studies is their reliance on static stimuli, which fails to capture the dynamic and active nature of human vision. To create a more ecologically valid paradigm, recent work by Solbach & Tsotsos (2023) introduced a fully 3D, interactive version of the task. Their setup used physical objects from the TEOS set Solbach & Tsotsos (2021), allowing participants to change their viewpoint and actively observe the stimuli. We replicate this modern setup in our simulation.

The TEOS objects are twelve 3D shapes inspired by Shepard and Metzler's originals, categorized into three complexity levels. In this task, "sameness" is defined as geometric congruence, and a shared coordinate system enables the precise quantification of the relative orientation (*RO*) between object pairs. Following the methodology of Solbach & Tsotsos (2023), we test our agents on three distinct *RO* values: $0°$, $90°$ and $180°$. Figure 2 illustrates the objects, *RO* values, and coordinate frame.

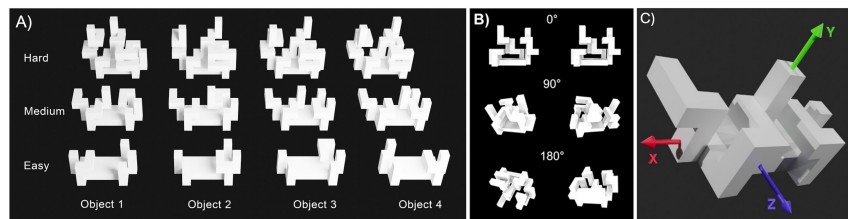

Figure 2: Stimuli used in this work. a) TEOS objects (from Solbach et al., 2021) categorized by three complexity levels based on block count. b) Illustration of the three *RO* values. c) Common coordinate system for all objects to define *RO*.

The human performance data from this task underscores its value as a benchmark. Although participants were highly accurate (93.8%), the problem proved non-trivial, requiring an average of 47.5 seconds, 16.6 meters of movement, and over 90 visual fixations to make a decision. Participants also demonstrated rapid learning, as their time and movement decreased over consecutive trials. This combination of a fundamental visuospatial challenge, the necessity for active exploration, and an observable learning curve makes it an ideal and demanding testbed for modern reinforcement learning agents.

## 3 LEARNING ENVIRONMENTS

We created a simulated environment in Unity using the ML-Agents toolkit Juliani et al. (2020) to replicate the human psychophysics experiment from Solbach & Tsotsos (2023). Using a simulation was necessary to overcome the inefficiency of real-world RL training, which often requires millions of trials. Our 3m x 4m virtual room (Figure 3) closely mimics the original setup's layout and lighting, with two target objects placed 1.2m apart at a height of 1.5m. A mobile agent, equipped

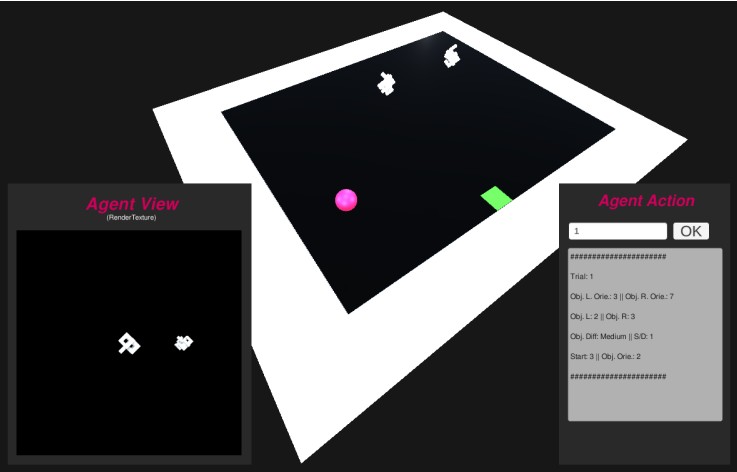

Figure 3: A screenshot of the Unity reinforcement learning environment where the agent (magenta sphere) interacts with two central objects. Insets show the agent's view (bottom left) and action history (bottom right). The green start square and white environment boundary are for illustration only and not visible to the agent.

with a single camera providing 512×512 pixel RGBA images as input, navigates this space to make a final binary "same" or "different" decision.

## 3.1 ACTION SPACE

The design of the action space is critical to a task's complexity. We investigated this by implementing seven distinct versions: one continuous and six discrete. The continuous space allows the agent to navigate freely, mimicking human movement, but results in an intractably large state space of over 500 million states[1], making learning difficult. To simplify the problem, our discrete action spaces restrict the agent to a predefined grid of viewpoints (6, 12, 24, 48, 72, or 96 locations, as shown in Figure 4). From each location, the agent's action is to orient its camera toward either of the two objects. This approach drastically reduces the state space to a manageable size (from 12 to 192 states), converting a complex navigation problem into a simpler selection task.

## 3.2 REWARD FUNCTION

We use a sparse reward function, provided only at the end of each trial, to encourage the agent to explore solutions without being overly guided Ecoffet et al. (2019); Lehman & Stanley (2011). The agent receives a terminal reward of +1 for a correct decision ("same" or "different") and -1 for an incorrect one. To promote valid behavior during the trial, a small penalty of -0.01 is applied under two conditions: if the agent is not looking at either of the target objects, or if it chooses a viewpoint outside the environment's boundaries.

## 4 THE REINFORCEMENT LEARNING SETUP

While supervised and unsupervised learning excel at finding patterns in static datasets, they are ill-suited for our interactive problem, which requires active data acquisition. We therefore use Reinforcement Learning (RL), where an agent learns a decision-making policy by interacting with an environment to maximize a cumulative reward. This paradigm aligns naturally with the exploratory nature of the Same-Different task. We evaluate four approaches: Proximal Policy Optimization (PPO) Schulman et al. (2017), Generative Adversarial Imitation Learning (GAIL), Behavioral Cloning Bain & Sammut (1995), and Curriculum Learning Bengio et al. (2009). The training hyperparameters for each method are provided in the supplementary material.

---

[1]Restricting the translation to $5cm$ at a time and orientation changes to $5°$.

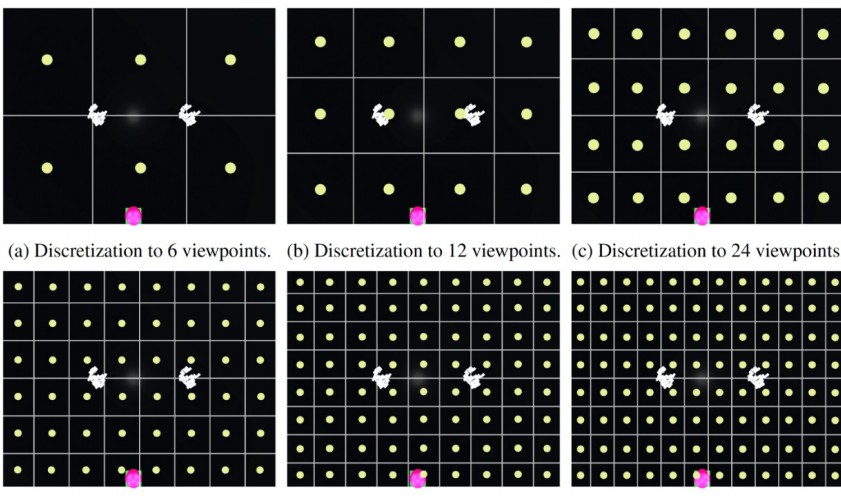

(a) Discretization to 6 viewpoints.  (b) Discretization to 12 viewpoints.  (c) Discretization to 24 viewpoints.

(d) Discretization to 48 viewpoints.  (e) Discretization to 72 viewpoints.  (f) Discretization to 96 viewpoints.

Figure 4: Illustration of the six top-down views (a-f) with different levels of action space discretization. The agent (magenta) can occupy any light green location and from there, can view either of the two white objects.

Proximal Policy Optimization (PPO) is a cornerstone of modern reinforcement learning, recognized for its robustness and state-of-the-art performance Schulman et al. (2017). Its effectiveness has been demonstrated across a wide range of complex challenges, from mastering strategic games like StarCraft II Vinyals et al. (2019) and solving intricate robotic manipulation puzzles Akkaya et al. (2019) to excelling in cooperative multi-agent benchmarks Yu et al. (2022). Given its proven versatility and success, PPO serves as a foundational algorithm in our study.

We also explore imitation learning, which leverages expert demonstrations to guide the agent. For this, we utilize a rich dataset of 791 human trials from Solbach & Tsotsos (2023), which is exclusively used for these methods. The first approach, Behavioral Cloning (BC), directly replicates the expert's policy by mapping observed states to the actions taken by the human subjects. While straightforward, BC can struggle to generalize to novel situations. The second method, Generative Adversarial Imitation Learning (GAIL), is a more robust technique that frames the learning problem adversarially. Based on GANs Goodfellow et al. (2020), GAIL trains a generator (the agent's policy) to produce behavior that a discriminator network cannot distinguish from the expert demonstrations. This allows the agent to learn a policy without an explicit reward function. For both BC and GAIL, we use PPO as the underlying training algorithm.

Finally, we also explored training the agent using curriculum learning with a PPO backbone. Curriculum learning divides the task into smaller subtasks (lessons) that are learned one after another going from simple to hard. We have designed two different lesson plans. The first, called the *naïve* lesson plan, trains the agent intuitively from easy to hard. Starting with easy object difficulty at *RO* 0°, then 90°, 180° (see Figure 2 B)), and finally all three *RO*, then moving to medium object difficulty and repeat the same *RO* sequence and so on (Table 1). The second, called *human experiment inspired* lesson plan, is based on the findings in Solbach & Tsotsos (2023). Solbach & Tsotsos (2023) reports human performance on this task, including their accuracy. We use the accuracy as an indication to determine the sequence of lessons. Interestingly, the order of lessons for medium and hard difficulties is different from the curriculum before. The order of exposing the agent to *RO* for medium and hard cases is now $90°, 0°, 180°$ instead of, and which seems more intuitive, $0°, 90°, 180°$ (Table 2).

## 5 EVALUATION

In this section, we present the results of learning the 3D Same-Different task using a variety of reinforcement learning frameworks. The training was stopped if the cumulative reward plateaued

| Lessons 1-4 | Lessons 5-8 | Lessons 9-13 |
|---|---|---|
| 1. Easy, $0°$ | 5. Medium, $0°$ | 9. Hard, $0°$ |
| 2. Easy, $90°$ | 6. Medium, $90°$ | 10. Hard, $90°$ |
| 3. Easy, $180°$ | 7. Medium, $180°$ | 11. Hard, $180°$ |
| 4. Easy, all *RO* | 8. Medium, all *RO* | 12. Hard, all *RO* |
| | | 13. all *RO* and difficulties |

Table 1: This table presents all 13 lessons. The lessons are chosen in an intuitive way from easy to hard. This is called the naïve curriculum

| Lessons 1-4 | Lessons 5-8 | Lessons 9-13 |
|---|---|---|
| 1. Easy, $0°$ | 5. Medium, $90°$ | 9. Hard, $90°$ |
| 2. Easy, $90°$ | 6. Medium, $180°$ | 10. Hard, $180°$ |
| 3. Easy, $180°$ | 7. Medium, $0°$ | 11. Hard, $0°$ |
| 4. Easy, all *RO* | 8. Medium, all *RO* | 12. Hard, all *RO* |
| | | 13. all *RO* and difficulties |

Table 2: This table presents all 13 lessons. The lessons are chosen based on the findings in Solbach & Tsotsos (2023). Note how the order of *RO* differs for medium and hard lessons.

for at least 5 million episodes. Here, an episode refers to one full trial as defined in Solbach & Tsotsos (2023), meaning an answer ("same" or "different") is provided. Each reinforcement learning configuration has been trained 12 times, and the reported results are averaged for all 12 models. If a model did not train the task due to bad random initialization Andrychowicz et al. (2020) or catastrophic forgetting McCloskey & Cohen (1989), it has not been included in this evaluation.

## 5.1 PROXIMAL POLICY OPTIMIZATION PERFORMANCE

Our first approach used PPO to learn the Same-Different task with randomized trials – no fixed order of *RO* or object difficulty. Figure 5 shows training results across setups: discrete environments (6 to 96 cells) and a continuous one. The x-axis lists these setups; the y-axis shows accuracy and average number of viewpoints, using a shared scale.

The agent only succeeded in the simplest, 6-cell environment, achieving 94.1% accuracy. Although it used an average of 5.65 viewpoints, it heavily favored a single viewpoint to make its decision. In all more complex setups, the agent failed to learn. For the 12 to 96 cell environments, accuracy hovered around 50%, indicating performance no better than random guessing. In the continuous environment, performance dropped to 0%, as the agent frequently failed to provide an answer at all.

## 5.2 IMITATION LEARNING PERFORMANCE

We trained two imitation learning frameworks, Behavioral Cloning (BC) and Generative Adversarial Imitation Learning (GAIL), on 791 human trials recorded in a continuous 3D space at 120 Hz Solbach & Tsotsos (2020). To use this data in our discrete environments, we mapped each human fixation to the nearest grid cell while preserving its original sequence and timing. This simplification ignored movements between fixations, direct agent control over $X, Y, Z$, and $pitch, yaw$, and assumed central object fixation.

Both frameworks failed to learn the task. BC was the least performant; across all 72 training attempts, its cumulative reward stagnated around 0.0 over 12 million episodes (Figure 6a). Similarly, no GAIL training attempt was successful. As shown in a representative run (Figure 6b), GAIL's training plateaued after 2.5 million episodes and was terminated.

## 5.3 CURRICULUM LEARNING PERFORMANCE

We trained an agent using curriculum learning (CL) with two different lesson plans, progressively increasing task difficulty. Figure 7 shows a representative training run for the 6-viewpoint discrete environment using the naïve lesson plan (Table 1), plotting the cumulative reward (a) and corresponding lesson progression (b).

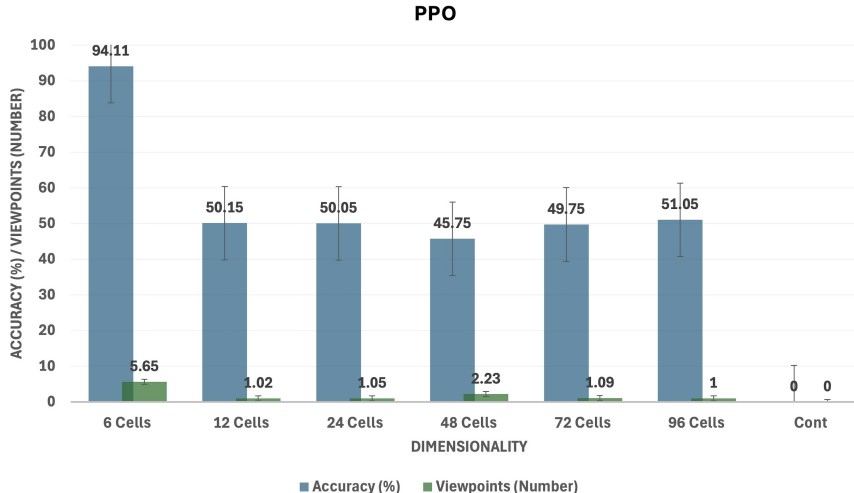

Figure 5: PPO agent training results: Average accuracy (blue) and viewpoints (green) across all environments.

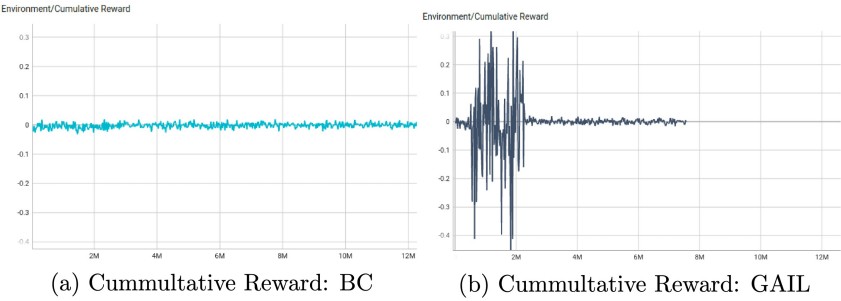

(a) Cummultative Reward: BC          (b) Cummultative Reward: GAIL

Figure 6: Example training result of BC (a) and GAIL (b) trained on the discretized environment using 48 possible viewpoints. Not showing any training effect over millions of episodes. In fact, the training plateaued.

Training was terminated after 18 million episodes at the onset of catastrophic forgetting McCloskey & Cohen (1989), where the cumulative reward suddenly dropped to zero. Performance had already begun to plateau around 14 million episodes, with the best of 12 training attempts achieving a 0.78 average reward. The agent learned initial lessons quickly but required significantly more time for harder configurations. We now examine the results from the naïve lesson plan across our seven setups, as shown in Figure 8.

The 6 viewpoint environment was learned at an accuracy of 95.88% on average, and the agent required 22.21 observations. Note that requesting a viewpoint from the same location is considered additional observations even though the input does not change. We will explore this in Section 6 in more detail. For the 12 viewpoint case, the performance improved to 97.9% on average with a reduction to 10.62 observations. Lastly, the performance of the environment with 24 cells was learned with 92.62% and required, on average, 16.27 observations. However, cases 48 through 96 are not learned as the accuracy is at 0%. We now examine the results from the lesson plan informed by human experiments Solbach & Tsotsos (2023) (Figure 9).

Under the human-informed plan, the agent successfully learned tasks with up to 48 viewpoints. It achieved high accuracies for the 6, 12, and 24 viewpoint scenarios (96.8%, 98.1%, and 93.81%). The 12-viewpoint model was the top performer, requiring only 5.11 fixations and, along with the 6-viewpoint model, surpassed average human accuracy Solbach & Tsotsos (2023). Performance declined for 48 viewpoints but was still considered a success (78.6% accuracy). However, the agent failed to learn the 72 and 96 viewpoint tasks ( 50% accuracy) and the continuous setup (0% accuracy). This curriculum learning approach was the most successful RL method tested.

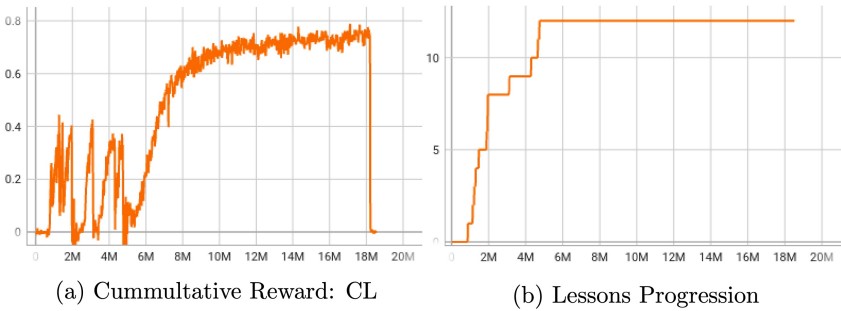

(a) Cummultative Reward: CL                    (b) Lessons Progression

Figure 7: Training insights of curriculum learning on the discrete environment with 6 viewpoints following the naïve lesson plan. a) Plot of the cumulative reward up to 18M training steps. b) The corresponding lesson progression.

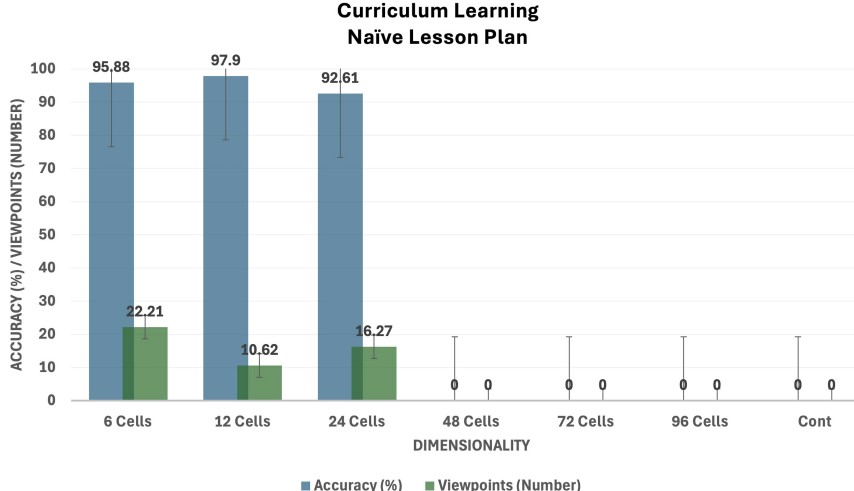

Figure 8: Performance of trained models (accuracy, fixations) was measured across six discrete and one continuous environment using the human experiment informed curriculum.

## 6 DISCUSSION

Our exploration of reinforcement learning frameworks revealed that "one-shot" methods like PPO and imitation learning struggled to learn across the entire problem space at once. In contrast, CL emerged as a successful strategy for differentiating the 3D objects, but its effectiveness was limited to environments with 48 or fewer viewpoints. This outcome underscores the potential of structured learning and points toward future work in developing frameworks that can master environments with larger action spaces.

The failure of imitation learning methods (BC and GAIL) is likely due to a significant semantic gap between the continuous human expert demonstrations and the simplified, discrete simulation. The restrictions of the training environment created a disconnect that the agents could not overcome. Conversely, the success of curriculum learning was greatly amplified when using a lesson plan informed by human data. This approach not only improved accuracy but, to our surprise, enabled the agent to solve a 48-viewpoint task that was previously unlearnable, validating the use of the human experiment data.

However, an analysis of the successful agent's strategy revealed a behavior starkly different from human vision. While humans purposefully select diverse fixations Solbach (2022) (i.e., they use their ability to attend to decide where to look and when), the CL agent learned to exploit a small set of dominant viewpoints, often repeatedly requesting the same observation. The best-performing model for the 12-viewpoint scenario, for example, chose a single location in 44.5% of its decisions (Figure

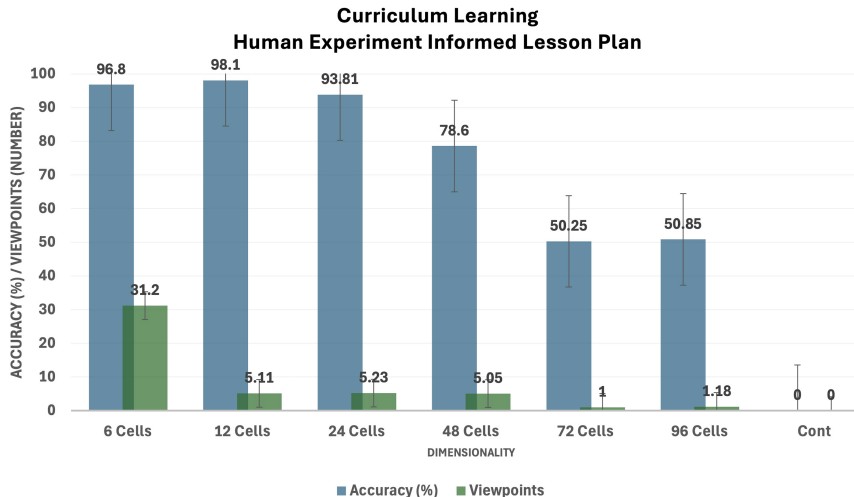

Figure 9: Performance of trained models (accuracy, fixations) was measured across six discrete and one continuous environment using the naïve curriculum.

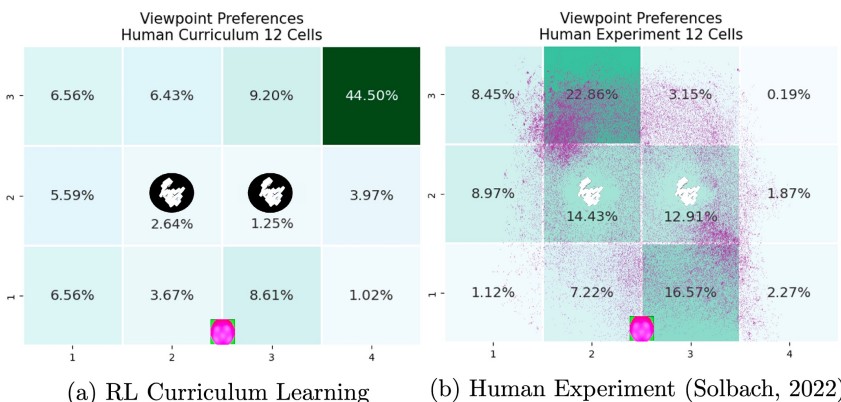

(a) RL Curriculum Learning    (b) Human Experiment (Solbach, 2022)

Figure 10: Heat maps comparing viewpoint distributions in a 12-cell grid. a) RL agent trained with a Human Curriculum b) Distribution of human fixations (with real fixation locations in purple).

10a). This reliance on a few key locations was a common characteristic across all successfully trained models.

In contrast, human subjects employ a planned, exploratory strategy, using a wide range of viewpoints and full head movement to investigate the objects Solbach (2022) (Figure 10b). The RL agent, however, adopted an exploitative strategy by concentrating its fixations on a single, information-rich area to quickly extract critical data. For example, Figure 11 shows a preferred agent viewpoint (a) that provides a decisive perspective on both objects (b).

The agent's learning progression showed a striking alignment with human cognitive patterns. It struggled most with lessons 9 and 10, which are the same tasks where human accuracy is known to decline Solbach & Tsotsos (2023). This suggests the agent is sensitive to the same sources of complexity that affect humans. While the agent found a workable strategy, humans solve this task with a much higher degree of efficiency and adaptability. Developing reinforcement learning agents that can achieve similar human-like performance presents a clear and valuable direction for future research.

One key aspect is the reliance on action history, evident in effective task strategies Solbach (2022) and a finding that contrasts with the approach taken by certain RL models. Humans often view objects sequentially, looking at one at a time. This behavior is only rational if they are remembering features from the first object to verify on the second, demonstrating a clear use of memory.

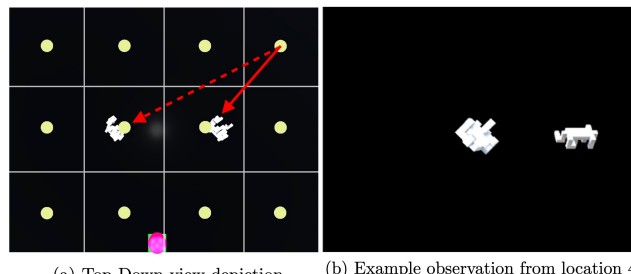

(a) Top-Down view depiction      (b) Example observation from location 4

Figure 11: An example observation from the 12-viewpoint environment. a) A top-down map showing the agent's focus (red arrow) and what it observes as a side effect (dashed arrow). b) The agent's resulting observation from the position in (a).

In contrast, an RL agent learned a different strategy: it finds viewpoints where both objects are clearly visible simultaneously, allowing for direct comparison (Figure 10 a)) – avoiding the need to memorize object details. Therefore, to capture the temporal dependencies and dynamic strategy adjustments that humans employ, RL methods require mechanisms for memory or history-based state representations.

Furthermore, the findings touch upon the practical application of reward-driven learning Silver et al. (2021). While reward is the fundamental signal, achieving learning in this sparse-reward setting required supplementing the primary objective with structured learning aids like curriculum learning. This suggests that efficiently solving complex tasks with RL often involves leveraging techniques beyond basic reward maximization to effectively guide exploration and strategy discovery.

Lastly, the exploration challenge inherent in this task highlights the practical limits of undirected trial-and-error. The need for agents to discover long, coordinated action sequences efficiently points towards the utility of more advanced RL methods. Techniques focusing on improved sample efficiency, such as guided exploration, hierarchical approaches, or integrating prior knowledge, are relevant directions for enabling RL to tackle problems of this scale and complexity.

## 7 CONCLUSION

This study investigated the application of modern reinforcement learning techniques to a complex, 3D Same-Different visuospatial task, drawing inspiration and benchmarks from human psychophysical experiments Solbach & Tsotsos (2023). Our exploration revealed that while state-of-the-art methods like PPO and imitation learning (BC, GAIL) encountered difficulties in directly acquiring optimal policies within this challenging, active perception environment, a structured approach using Curriculum Learning proved effective. By designing a curriculum informed by human performance data, we successfully trained agents capable of solving the task, albeit employing strategies (such as viewpoint preferences) that differed distinctly from human observers. This is notable, as neural network learning techniques have also demonstrated difficulties on 2D versions of a similar task. For instance, Kim et al. (2018) found that feedforward neural networks are strained by same-different relationship tasks, struggling to generalize beyond rote memorization when faced with stimulus variability.

The success hinged on simplifying the problem initially and progressively increasing complexity, highlighting the critical role of guided learning and exploration in sparse-reward, high-dimensional state spaces typical of realistic visuospatial problems. These findings suggest that while RL holds promise for tackling complex cognitive tasks, achieving robust, human-like performance may necessitate integrating structured learning paradigms like curriculum learning or incorporating cognitive principles such as attention, memory, or explicit planning mechanisms to be able to cope with action spaces beyond 48 cells. Future work should focus on developing such cognitively-inspired RL architectures, using challenging benchmarks like the 3D Same-Different task to drive progress towards more generally capable and adaptable intelligent agents.

ACKNOWLEDGMENTS

*REMOVED FOR DOUBLE-BLINDNESS*

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

## A    APPENDIX

## B    HYPERPARAMETER LISTS

Hyperparameters used to train various RL algorithms are presented here.

### B.1    PPO

To set up PPO, we chose the following hyperparameters as listed in Table 3. In dozens of training rounds, we have carefully tuned these empirically.

| Hyperparam. | Value | Meaning |
|---|---|---|
| Network | Simple CNN | Which neural network is used for policy training. |
| Layers | 2 | How many hidden layers are present in the model. |
| Hidden Units | 256 | How many hidden units are in each fully connected layer |
| Batch Size | 128 | How many experiences (observations, actions, rewards) are used for one iteration of a gradient descent update. |
| Beta | 0.005 | Strength of the entropy regularization |
| Epsilon | 0.2 | Acceptable threshold between old and new policies. |
| Lambda | 0.95 | Parameter for GAE (Generalized Advantage Estimate). Prioritization of the current value estimate when calculating and updated value estimate. |
| Buffer Size | 1024 | How many experiences are collected before updating the model. |
| Gamma | 0.99 | Discount factor for future rewards. |
| Strength | 1.0 | The weight by which the raw reward is multiplied. |
| Learning Rate | 3.0e-4 | Strength of the gradient descent update step. |
| Max Steps | $10^8$ | The number of simulations are run for the entire training process before termination. |

Table 3: PPO's Hyperparameter using Unity's ML-Agents.

### B.2    BC

We train BC using PPO as its backbone. The parameters of PPO are the same as listed in Table 3. Table 4 shows the hyperparameters that are specific to behavioural cloning.

| Hyperparam. | Value | Meaning |
|---|---|---|
| Strength | 0.5 | The learning rate of behavioral cloning relative to PPO. |
| Steps | 0 | Number of training steps over which behavioral cloning is active. 0 means constant imitation over the entire training run. |
| Batch Size | 128 | The amount of demonstration experience used for one iteration of a gradient descent update. |
| Num Epoch | 5 | The number of passes through the experience buffer during gradient descent. |
| Sample per Update | 0 | How many demonstrations are used for an update step. 0 means to use all demonstrations. |

Table 4: Behavioral Cloning's Hyperparameters using Unity's ML-Agents.

## B.3 GAIL

Table 5 shows the hyperparameters that are specific to GAIL.

| Hyperparam. | Value | Meaning |
|---|---|---|
| Strength | 1.0 | The learning rate of GAIL relative to PPO. |
| Gamma | 0.99 | Discount factor for future rewards. |
| Learning Rate | 128 | Learning rate used to update the discriminator |
| Hidden Units (Discriminator) | 128 | How many hidden units are in each fully connected layer. |
| Layers (Discriminator) | 2 | How many hidden layers are present in the model. |
| Network (Discriminator) | Simple CNN | Which neural network is used for policy training. |
| Use Actions | False | Whether the discriminator uses both observations and actions or just observations. False means that the agent visits the same states as in the demonstrations but allows for different actions. |
| Use Vail | False | This sets the variational bottleneck within the discriminator. |

Table 5: GAIL's Hyperparameter using Unity's ML-Agents.

