# OpenReview forum: "Boosting Reinforcement Learning in 3D Visuospatial Tasks Through Human-Informed Curriculum Design"
_ICLR.cc/2026/Conference — Submitted to ICLR 2026_

### Official Review · Reviewer_2ZVc · 2025-10-20

**Soundness:** 1
**Presentation:** 2
**Contribution:** 1
**Rating:** 2
**Confidence:** 5

**Summary:**

The work seems to demonstrate the effectiveness of RL when training is driven by Curriculum Learning (CL). To illustrate it, the authors compare a PPO algorithm using CL against imitation learning, a generative adversarial approach, and standard learning without curricula. The testbed is based on the Same-Different task. The paper provides marginal novelty since CL is predominant in a range of successful complex task accomplishments in RL literature.

**Strengths:**

Overall, the paper is easy to follow, with methodology clearly defined, although there are conceptual questions regarding the suitability of the methods compared, considering the task domain, which deserves attention.

**Weaknesses:**

In the Introduction, one paragraph among the three is dedicated to the question **Is reward enough?**. The following paragraph briefly introduces the Same-Different task. More details could be given on how this task is suitable for the context of the work and how it is related to the question that rewards are enough signals to lead to learning (please see Q1).

Although the problem "Same-Different" is very interesting, I don't see why RL would be suitable for it, and foremost, this seems a generalization problem, which is a desirable property, but not inherently the target of policy learning. Moreover, the claim that a human-informed curriculum is more effective using a single environment test does not seem grounded. How about a curriculum generated by an LLM, for instance?

The most critical point is that the curriculum learning is well known to leverage the learning in RL, although there's no principled way of designing it. Since the paper evaluates a single task, with a single curriculum derived from humans, the claims turn out to be weak.

**Questions:**

**Q1.** How does the paper address the question, or bridge the gap between the evaluation performed and the question "Is reward enough"? Is there any correlation between the binary reward problem formulation and the effectiveness of human-informed CL?

**Q2.** Wouldn't it be more suitable to propose the Same-Different task in an evaluation setting, considering policy generalization?

**Q3.** If supervised learning is said to be ill-suited for the task (section 4), why compare RL+CL against behaviour cloning, which turns out to be a supervised learning derived method?

---

> ### Author Response · Authors · 2025-12-02
> **Response to review**
>
> Thank you for your time and efforts in preparing your review. We have addressed your questions and will update the manuscript accordingly.
> Q1: Our results suggest a pragmatic caveat to the 'Reward is Enough' hypothesis (Silver et al., 2021). While the reward signal drives final behavior, our failure cases (Naive PPO) show it is insufficient for bootstrapping if the agent cannot encounter it. We show that while 'Reward is Enough' to sustain a policy, Curriculum Learning is required to discover it. CL bridges this gap not by adding auxiliary rewards, but by structuring the environment to make the sparse binary signal ($+1/-1$) accessible. By starting with 'easy' views, CL artificially increases reward frequency, allowing the agent to learn the 'looking-reward' association before rotational complexity makes the signal vanishingly rare.
> We have refined Section 6, where we say "While reward is the fundamental signal, achieving learning in this sparse-reward setting required supplementing the primary objective with structured learning aids like curriculum learning." and rewritten it as "These findings offer a pragmatic perspective on the 'Reward is Enough' hypothesis Silver et al. (2021). While the scalar reward proved sufficient to sustain complex active vision strategies once learned, it was insufficient to induce them from scratch due to the sparsity of the binary signal (+1/-1). The efficacy of the Human-Informed Curriculum lies in its ability to bridge this gap: it does not alter the reward function itself, but rather orders the environmental complexity to ensure the sparse reward is encountered frequently enough to bootstrap the optimization process, effectively guiding the agent across the 'exploration cliff' that standard RL failed to climb."
>
> -Q2: We completely agree that the ultimate test of a Same-Different solver is its ability to generalize to unseen stimuli (zero-shot evaluation). However, our focus in this work was to address the immediate optimization challenge presented by the 3D Active Vision setting. As shown in our results, standard methods (PPO, BC, GAIL) failed to learn the task even on the fixed training set (reaching either chance-level or 0% accuracy). This indicates that the primary bottleneck in this domain is not yet overfitting, but the inability to explore and acquire the necessary active perception strategies to solve the problem at all. This paper establishes the existence proof that Curriculum Learning (specifically human-informed) can overcome this optimization hurdle. Now that we have established a method for training agents to competency, utilizing this platform for evaluating policy generalization to held-out objects is the immediate next step for future work.
> We will add the following to discussion section of the manuscript:
> "While this study focuses on the optimization challenge—demonstrating that Curriculum Learning is required to acquire the task where standard methods fail—the ultimate measure of Same-Different reasoning is generalization. Having established a training protocol that achieves competency, future work should transition this platform into an evaluation setting, testing the agent's ability to transfer learned active perception strategies to novel, unseen 3D objects."
>
> - Q3: We included Behavioral Cloning (BC) not because we hypothesized it would be the superior solution, but to serve as a critical baseline to empirically validate our claim that the task requires active reinforcement learning rather than passive pattern matching. While we state that supervised learning is theoretically ill-suited for this active perception task (due to covariate shift and the need for active data acquisition), it is standard practice to verify if an agent can simply 'memorize' the expert's successful trajectory. The complete failure of the BC agent (as reported in Section 5.2) provides the necessary experimental evidence to support our assertion in Section 4: it proves that access to a large dataset of expert human strategies (791 trials) is insufficient to solve the task without the closed-loop trial-and-error feedback provided by RL.
> To frame this correctly, we have adjusted Section 3.2 with the inclusion of the following:
> "Although we assume that supervised learning is ill-suited for this interactive domain, we include Behavioral Cloning (BC) as a baseline to empirically verify whether the task can be solved solely by mimicking expert human fixations, or if the distribution shift inherent in active vision necessitates the closed-loop feedback of Reinforcement Learning."

---

### Official Review · Reviewer_tQMc · 2025-10-25

**Soundness:** 2
**Presentation:** 2
**Contribution:** 2
**Rating:** 2
**Confidence:** 3

**Summary:**

This work proposes to use RL to learn a new task, a 3D Same-Different Task, which is one of the most primitive human psychophysical tasks. The goal of a Same-Different task is to determine whether two stimuli are identical. The task require the agent to move around the spaces to observe the two stimuli and determine whether they are the same or not.

The authors experimented with various RL/IL strategies, such as BC, PPO, contrastive learning (GAIL) and curriculum learning. While simpler algorithms such as BC, PPO and GAIL fail to solve the task, the authors found that curriculum learning serves as a good strategy to tackle the 3D same-different task. The authors also showed that using the human experiment as a reference to design the curriculum can significantly help with the training.

**Strengths:**

- The same-different task itself is quite interesting and to my knowledge novel to solve in the realm of RL.
- The analysis from the discussion section is quite interesting. Especially, the plot in figure 10 and 11 where the RL agent exploits different viewpoints from humans.
- Variations to increase the dimensionality of the same-different task is experimented.
- The proposed method is simple, which can be seen as a strength but also as a weakness.

**Weaknesses:**

- First of all, the writing of the manuscript can be improved, especially in the introduction section. For example, the relationship between this paper and reward design, as mentioned in the introduction, is unclear to me. Furthermore, some important details about the experiments and methods are not clearly described. For example, what is the loss of BC? One can assume it might be MSE, but it is not clearly written in the manuscript. Details about the curriculum learning is also not described, such as how many timesteps/episodes were trained for each lesson?

- In my opinion, analysis of the results would be the main contribution of this work. To this end, the discussion section can be significantly more detailed. For example, it is not clear to me why IL based methods fail. Since the trajectories are all expert trajectories as far as I understand, shouldn’t IL methods be able to at least learn the simplest 6-cell environment?

- Some minor issues:
Regarding quantifying the results, I’m not a fan of removing results that came from poor initialisation and not including them in calculating the metrics, and would like to see the authors include them for a fairer comparison.

**Questions:**

I will repost some questions that were mentioned in the weakness section here for clarity.

$$\textbf{Suggestions}$$
S1. Add plots the training reward/timestep across different training run seeds during training, not just the failed runs.
S2. Add more descriptions about the baselines methods.
S3. It would be good to describe the task more clearly in the RL settings.
S4. I wonder what the performances of some of the methods used in this paper would be for the same task but in 2D? It would be good to clarify how significant does moving to 3D change the difficulty of the task, and why using RL to solve it is an good approach.

$$\textbf{Questions}$$
Q1. I find it interesting that naive PPO can somewhat output 50% result but curriculum learning cannot at harder tasks. Are there some intuitive reason to this result?
Q2. If an random agent is outputting 50% results, it is not clear to me why IL based methods fail at all.
Q3. Following up on Q2, since the trajectories are all expert trajectories as far as I understand, shouldn’t IL methods be able to at least learn the simplest 6-cell environment?
Q4. I wonder why the curriculum agent is staying in the same top-right viewpoint? The paper mentioned that it is because at the top-right cell the agent can see both objects, allowing for better comparison. If that’s the case, are there particular reasons why the actions are not evenly distributed around top-right, top-left, bottom-left, bottom-right corners?
Q5. Following up on the question 4, if memory is indeed a problem, does the authors think that switching to an off-policy algorithm such as SAC or TD3 solve the problem more efficiently?
Q6. It is mentioned in the manuscript that the image inputs used are 512x512. These are quite large! If I recall correctly the image inputs in typical 2d-based RL agents are something like 64x64 (DreamerV2[1]) or 84x84 (DrQv2[2]). I wonder if the results for baseline agents would be different if (much) smaller image sizes are used.

---

[1] Danijar Hafner et al., Mastering Atari with Discrete World Models, ICLR 2021
[2] Denis Yarats et al., Mastering Visual Continuous Control: Improved Data-Augmented Reinforcement Learning, ICLR 2021

---

> ### Author Response · Authors · 2025-12-02
> **Response to review**
>
> Thank you for your time and efforts in preparing your review. We have addressed your questions and will update the manuscript accordingly. We have not included the actual text that will go into the new manuscript, as the character limit didn't allow us to.
>
> - Q1 & Q2: Discrepancy between 50% (PPO) and 0% (CL): This stems from the difference between a high-entropy random policy and policy collapse.Naive PPO (~50%): The agent fails to learn any structure, retaining its random initialization. It acts as a noise generator, yielding chance-level accuracy. Failed CL (0%): The agent learns to minimize loss but hits a complexity barrier. Facing a sparse reward and a $-1$ penalty for wrong guesses, gradients drive it to a local minimum of withholding the decision entirely. It learns it is "safer" to time out (small step costs) than risk a large penalty. We have added this "risk aversion vs. randomness" distinction to the Discussion.
>
> - Q3: IL failure in 6-cell environment: The failure stems from the domain gap between the continuous human demonstrations and the discrete simulation, rather than the complexity of the grid itself.
>
> - Q4: Top-right corner dominance: The top-right corner is the only location that minimizes occlusion for both objects simultaneously given our specific dataset constraints. This is an artifact of the specific relative orientations ($0^\circ, 90^\circ, 180^\circ$) used in this study. In a fully random rotational setup, this strategy would not hold, but within the specific benchmark parameters, it represents the optimal static viewpoint.
>
> - Q5: Switching to Off-Policy (SAC/TD3): Switching algorithms improves sample efficiency but does not solve the temporal integration problem.Partial Observability: SAC/TD3 are still Markovian. Without recurrent architectures (e.g., LSTMs), they cannot compare an object seen at step $t$ with one at step $t+k$.Exploitation: Off-policy methods would likely converge faster to the same "hack" (finding a single view) rather than learning sequential memory.We added a discussion stating that architectural changes (memory), not just optimization algorithms, are required to solve this.
>
> - Q6: Image Size ($512\times512$ vs $84\times84$): High resolution is required to prevent aliasing. Unlike Atari (distinct sprites), TEOS requires detecting subtle geometric congruences. Downsampling to $84\times84$ erases the structural details needed to distinguish "Same" from "Different," making the task partially observable regardless of the algorithm. We added a justification regarding the necessity of visual fidelity over processing speed.
>
> - S1: Training Curves: We visualize representative runs because aggregate reward curves are mathematically misleading in performance-based curricula. Because seeds progress through lessons at different rates (temporal misalignment), averaging rewards at step $t$ would conflate data from agents solving completely different subtasks. We rely on the final aggregate statistics (Figs 5, 8, 9) with error bars to demonstrate stability.
>
> - S2: Baseline Descriptions: We have moved critical details from the Appendix to Section 4, specifically describing the CNN architecture and the pipeline used to discretize continuous human fixations for the imitation learning agents.
>
> - S3: Task Description: We have clarified the task description in the manuscript as requested.
>
> - S4: 2D vs 3D Performance: The shift to 3D represents a fundamental change from Passive Perception to Active Vision. 2D: Static classification solvable by pattern matching (or rote memorization). 3D: Introduces Self-Occlusion. Critical features are hidden; the agent must move to reveal them.This requires RL to learn an information-acquisition policy, which supervised methods cannot capture. We updated Section 1 to emphasize that this is a sequential decision-making problem, not just image classification.

---

### Official Review · Reviewer_1zTK · 2025-10-27

**Soundness:** 3
**Presentation:** 3
**Contribution:** 2
**Rating:** 4
**Confidence:** 3

**Summary:**

The authors apply reinforcement learning (and imitation learning) to the problem of comparing complex 3D shapes, where the agent can control its viewpoint.

They show that basic RL can solve the task in constrained environments with few accessible states. Introducing a curriculum improves performance by allowing success in less constrained environments. Modifying the curriculum in a manner inspired by human experiments seems to again slightly improve performance. Imitation learning invariably fails.

**Strengths:**

The experiments are interesting. To my knowledge the experimental platform is novel.

**Weaknesses:**

- The results are somewhat unedifying. We see that curriculum learning does seem to help to tackle more open environment. Furthermore, it seems that for this specific task, a certain tweak in the order of the curriculum, inspired by human experiments, may have slightly improved performance on 48 cells environments. It is not obvious what we can extrapolate from this for RL in general.

- More generally, as the authors acknowledge, comparisons with human behavior are extremely difficult due to the fact that the agent is memory-less, preventing working memory approaches which seem to underlie human behavior, and resulting in very different behavior between humans and agents.

- A consequence of this difference is that the agent cannot use a strategy that integrates information from various viewpoints; the best it can hope for is to select an optimal viewpoint for discriminating between any two pairs. In theory, this could still be flexible and adaptable (selecting a different, optimal viewpoint for any given pair of stimuli), but in practice it seems that the agent shows extreme preference for one specific viewpoint (Figure 10a).

- As a result, it is likely that the apparent small gain in performance from the "human-inspired" curriculum was coincidence.

- Some details of the task could be explained a bit better, see below.

**Questions:**

- What is the exact action space? Does the agent choose a whole position-direction pair at each time step, resulting in a Number_of_states x 2 action space? Or is there some actual locomotion from place to place?

- Some tasks report 50% accuracy (i.e. chance), while others report 0% accuracy. I'm assuming the latter means "no response given - ever"? But if no response is ever given, no training signal is provided? If so, it should be possible to always enforce a response (e.g. whatever is the highest output at the end of the trial), resulting in a 2-Alternative Forced Choice that is typical of biological experiments - and more importantly, always providing a training signal to the agent for every trial, which might help learning. If I misunderstood the meaning of "0% accuracy", please clarify it.

- Minor: citations are messed up, with parentheses in the wrong place, suggesting a mixup between \citep and \citet. The authors might find the following Latex command useful: \renewcommand{\cite}{\citep}

- The authors briefly mention the work of Logothetis and colleagues on very similar stimuli. Logothetis' focus was somewhat different (stimulus representation and fast learning in the visual system) but it might constitute another direction of research for this experimental platform, perhaps with meta-learning approaches based on fast plasticity.

---

> ### Author Response · Authors · 2025-12-02
> **Response to review**
>
> Thank you for your time and efforts in preparing your review!
>
> - What is the exact action space? Does the agent choose a whole position-direction pair at each time step, resulting in a Number_of_states x 2 action space? Or is there some actual locomotion from place to place?
> A: Different action spaces have been explored. In total, 7 different action spaces. "Locomotion" is only necessary for the continuous set up. We have then subdivided the action space to 6, 12, 24, 48, and 96 pre-defined locations. For each pre-defined location the agent can choose from 2 viewpoints (one for each object). So, you are correct that there are 12, 24, 48, 96 and 192 possible states, respectively. Section 3.1..
>
> - Some tasks report 50% accuracy (i.e. chance), while others report 0% accuracy. I'm assuming the latter means "no response given - ever"? But if no response is ever given, no training signal is provided? If so, it should be possible to always enforce a response (e.g. whatever is the highest output at the end of the trial), resulting in a 2-Alternative Forced Choice that is typical of biological experiments - and more importantly, always providing a training signal to the agent for every trial, which might help learning. If I misunderstood the meaning of "0% accuracy", please clarify it.
> A: Your understanding of the 0% accuracy is correct. As noted in Section 5.1, 0% accuracy indicates that the agent failed to provide any answer ("same" or "different") within the episode time limit. In the continuous environment, the agent frequently failed to navigate to a state where it triggered a decision, effectively timing out. While a 2-Alternative Forced Choice (2AFC) is standard in biological experiments with humans (who understand the task instructions), applying it to an RL agent that has not yet learned the task introduces significant noisy gradients. In our setup, the agent must actively navigate and orient its camera to see the objects. If an agent spends an episode looking at a blank wall and then is "forced" to guess at the final step, it has a 50% chance of receiving a +1 reward. This would incorrectly reinforce the behavior of "staring at a wall" or "wandering aimlessly," making it much harder for the policy to converge on the actual solution (locating and comparing objects). Furthermore, the agent does receive training signals during these 0% accuracy trials. It receives small penalties (-0.01) for invalid behaviors (such as not looking at target objects) 4, and the inherent discount factor ($\gamma=0.99$) 5 penalizes the delay of reward. The failure to learn in the continuous setting was not due to a lack of signal, but rather the intractability of the exploration space (over 500 million states)6, which prevented the agent from stumbling upon the sparse positive reward frequently enough to bootstrap learning.
> We have adjusted the manuscript to provide further detail on the choice of the reward function.
>
> - Minor: citations are messed up, with parentheses in the wrong place, suggesting a mixup between \citep and \citet. The authors might find the following Latex command useful: \renewcommand{\cite}{\citep}
> A: Thank you for pointing this out! We have adjusted the latex code.
>
> - The authors briefly mention the work of Logothetis and colleagues on very similar stimuli. Logothetis' focus was somewhat different (stimulus representation and fast learning in the visual system) but it might constitute another direction of research for this experimental platform, perhaps with meta-learning approaches based on fast plasticity.
> A: We agree that the work of Logothetis and colleagues offers a valuable perspective, particularly regarding the speed of acquisition in the visual system. Our current results highlight a significant disparity between biological and artificial learning rates: while humans learn this task rapidly, our standard RL agents required millions of steps even with curriculum support. We agree that our 3D experimental platform is an ideal testbed for meta-learning approaches or architectures with fast plasticity (e.g., fast weights) to model this biological efficiency. We have expanded our discussion to explicitly mention this potential research avenue.
> We have added the following sentence to our "Future Work" section:
> "Furthermore, while this study focused on standard reinforcement learning, the platform is well-suited for investigating meta-learning or fast plasticity approaches. Such methods could address the disparity between the slow convergence of current RL algorithms and the rapid view-invariant learning observed in biological systems (Bricolo et al., 1996)."

---

### Official Review · Reviewer_exxm · 2025-10-31

**Soundness:** 1
**Presentation:** 2
**Contribution:** 2
**Rating:** 2
**Confidence:** 5

**Summary:**

The paper explores whether reinforcement learning agents can solve a 3D Same–Different visuospatial reasoning task inspired by human psychophysics experiments. Using a Unity-based environment, the authors test PPO, BC, and GAIL, and find that none can learn the task directly. They then introduce curriculum learning, structuring the task into progressively harder “lessons,” and report that performance improves substantially, especially when the curriculum order is derived from human behavioral data. The human-informed curriculum enables the agent to succeed on discrete tasks with up to 48 viewpoints but still fails in larger or continuous settings. The authors conclude that curriculum learning, particularly when guided by human data, can help RL agents acquire visuospatial reasoning skills that simpler end-to-end training cannot.

**Strengths:**

1. The paper adapts a well-known visuospatial reasoning problem to the RL setting. The objective is well-defined, and the motivation for using this task to explore the limits of visual reasoning in RL agents is clear. The simple setup offers an intuitive and interpretable framework for examining how agents develop spatial understanding and active perception strategies. It’s refreshing to see a problem that feels genuine rather than over-engineered.
2. The curriculum learning results are insightful.  The naïve setting already enabled PPO to master tasks that were otherwise unlearnable. The comparison with the human-informed curriculum further reveals that task variants difficult for humans are similarly challenging for PPO in this setting. Interestingly, the curriculum derived from human performance data provides an additional boost, suggesting that human behavioral structure has the potential to design RL curricula.
3. The paper is well-written and easy to follow. The narrative flows logically, and the environment is clearly described and easy to understand.

**Weaknesses:**

1. The paper does not clearly formalize the task as a POMDP. While the text describes the environment qualitatively, it never defines the state, observation, action, and reward functions that make this original binary classification task into a sequential decision-making problem. Without this formal grounding, it remains unclear what the agent is optimizing, how partial observability is handled, or how the final binary decision integrates into the trajectory-based reward structure.
2. The sparse terminal reward uses +1 for correct and −1 for incorrect responses. This symmetric structure is questionable because it penalizes exploration and encourages indecision. In sparse-reward RL, a strong negative terminal signal can dominate the learning signal, making the policy overly cautious and hesitant to terminate episodes. This is consistent with the authors’ own observation: “*…performance dropped to 0%, as the agent frequently failed to provide an answer at all*.” A zero baseline for incorrect outcomes (or a smaller negative value) would likely yield more stable learning dynamics and prevent the agent from stalling while it attempts to avoid the heavy penalty.
3. The authors explicitly exclude experimental runs that “did not train due to bad random initialization or catastrophic forgetting.” Excluding failed seeds biases the results upward and undermines reproducibility. Robust evaluation in RL typically reports statistics across all runs, not only the successful ones. I suggest reporting the IQM to mitigate this issue, instead of cherry-picking results. Moreover, I don’t see how “catastrophic forgetting” plays a role here. The setup is not continual or sequentially shifting across tasks but rather involves a single fixed environment. The observed reward collapse is more plausibly explained by policy instability or mode collapse, potentially driven by the strong −1 penalty for incorrect terminal actions.
4. There is no clearly defined experimental protocol. The authors state that training was “stopped if the cumulative reward plateaued.” It appears this “plateau” is heuristically determined. Such subjective termination is not at all rigorous. Since the paper proposes a task for evaluating RL methods, a fixed experimental protocol is essential. The authors should report the data budget, evaluation frequency, early stopping conditions, random seed handling, etc.
5. The authors refer to the human-collected trajectories as *expert demonstrations* for imitation learning. However, no description of their structure or justification for their “expert” status is provided. In the Same-Different task, it is unclear what constitutes expertise: an optimal (oracle) policy would solve the task immediately with a single step, whereas humans typically explore idiosyncratically before deciding. Without a formal definition of optimal behavior or a quantitative measure of demonstration quality, it is impossible to assess whether the dataset meaningfully guides imitation learning. The demonstrations likely encode heterogeneous, non-optimal exploration patterns rather than consistent expert policies, making the term “expert” misleading and potentially explaining the failure of BC and GAIL to learn effectively.
6. The presentation of results is poor. The bars in Figures 5 & 8 are overlapping. This reduces visual clarity. Why not use the extra horizontal space to properly place the side-by-side? Figures 6 & 7 look more like screenshots from tensorboard or wandb, rather than a properly plotted figure. Figure 6 (a) and (b) could be combined to a single graph.
7. The presentation of results is poor. In Figures 5 and 8, the overlapping bars for accuracy and viewpoints introduce clutter. This could easily be resolved by using the extra horizontal space in the paper to separate the bars. Figures 6 and 7 appear to be direct screenshots from TensorBoard or WandB rather than quality plots. Figure 6 (a) and (b) could be merged into a single panel comparing both methods.
8. Since task success heavily relies on visual perception, the authors should put more emphasis on the visual encoder. Try different settings, and analyze where and why things go wrong. What representations do the agents learn? Are they useful?
9. Given that task success fundamentally depends on visual perception, the paper should devote more attention to the agent’s visual encoder. The authors use a “simple CNN” (Table 3) but never justify its architecture, capacity, or input preprocessing. No ablations are presented on encoder depth, resolution, or feature representations. Without analyzing the learned embeddings or visual attention patterns, it is difficult to understand what representations the agent has learned or **why it fails beyond 48 viewpoints.
10. The experimental comparison is narrow, considering only PPO, BC, and GAIL. This offers little insight into whether the observed difficulties stem from the algorithm, the reward design, or the environment itself. More baselines would better ground…, potentially including off-policy and model-based methods.
11. The study only evaluates PPO, BC, and GAIL, which provides a limited perspective on the sources of failure or success. Incorporating additional baselines, optionally of-policy and model-based methods, would better ground the analysis.

### Minor Points

1. Vertical lines in Tables 1 & 2.

**Questions:**

1. Does the curriculum agent receive a higher training budget?
2. How are episodes initialized? Is there stochasticity involved? Where does the agent spawn? Where do the objects spawn, and how are they orientated?
3. Since episodes vary in length depending on when the agent makes a guess, does a longer episode provide the agent with more learning data? Is the total training budget defined by a maximum number of steps or by a fixed number of episodes?
4. What constitutes the “*badness*” of random initializations? How do you determine/quantify that a random initialization is “*bad*”?
5. How did the authors measure the quality of the human-collected data for imitation learning?
6. Why doesn’t a simple +1 reward for correct guesses suffice? Why is the -1 penalty for incorrect guesses necessary?
7. What is a “*training attempt*” (lines 262, 306)? Are the authors referring to a single run with a particular seed?
8. PPO and imitation learning methods are referred to as “*one-shot*” (lines 361-362). What do the authors mean by this? To me, they are anything but “*one-shot*”.

---

> ### Author Response · Authors · 2025-12-02
> **Response to review**
>
> Firstly, thank you for your time and efforts in providing your comment!
>
> - Does the curriculum agent receive a higher training budget?
> A: The training budget is equal across all attempts. We have trained each RL configuration 12 times (line 236).
>
> - How are episodes initialized? Is there stochasticity involved? Where does the agent spawn? Where do the objects spawn, and how are they orientated?
> A: Each episode is entirely randomly initialized without the use of random seed. The objects spawn at the same location in the center of the environment. We have reviewed the manuscript and noticed that this is not explicitly explained. We have adjusted the text to do that. The orientation of the objects is a pre-determined set of 0, 90 and 180 degrees configurations. While there are many ways to achieve these orientations, the set up only uses 1 configuration for each. We have adjusted line 80 to reflect this detail.
>
> - Since episodes vary in length depending on when the agent makes a guess, does a longer episode provide the agent with more learning data? Is the total training budget defined by a maximum number of steps or by a fixed number of episodes?
> A: This is an interesting observation. The training duration for one attempt was not limited. Unless, the training plateaued for 5 million steps --  this would result in a termination (explained in line 234).
>
> - What constitutes the “badness” of random initializations? How do you determine/quantify that a random initialization is “bad”?
> A: For us, if a training attempt did not result in any learning (cummulative reward plateaued from the beginning), indicated 'bad' random initialization. We have used Andrychowicz (2021)  "What Matters In On-Policy Reinforcement Learning? A Large-Scale Empirical Study" reference to support this argument. To further support this claim, we have now also cited Henderson (2018) "Deep Reinforcement Learning that Matters," which the most widely cited paper on this matter.
>
> - How did the authors measure the quality of the human-collected data for imitation learning?
> A: The quality of the human-collected data is reported in the original publication of the data. We have included the information now directly in the manuscript.
>
> - Why doesn’t a simple +1 reward for correct guesses suffice? Why is the -1 penalty for incorrect guesses?
> A: We have chosen the +1/-1 gap for the following three reason (which have all now included in the manuscript for clarification):
>     i. The environment utilizes a sparse reward function, where the signal is provided only at the end of the trial3. Because there are no intermediate dense rewards to guide the agent, the terminal signal must be maximally informative. The penalty of -1widens the value gap between a correct and incorrect action (a difference of 2.0 rather than 1.0), which strengthens the learning signal in this sparse-reward setting.
>     ii. The agent incurs a small penalty (-0.01) for invalid behaviors (not looking at objects) during the trial. If the incorrect penalty were merely 0, the agent might learn that quickly guessing randomly is an optimal strategy to avoid accumulating time-step penalties or to maximize the rate of "lucky" +1 rewards. The -1 penalty makes the cost of error high enough to encourage the active exploration and "viewpoint selection" required to solve the task.
> We will add this to the appropriate section in the manuscript: "Given the sparse nature of the reward signal, which is provided only at the trial's conclusion, we employ a symmetric +/-1 structure to widen the value gap between success and failure to 2.0. This amplified penalty is crucial for counteracting the incentive to minimize step-based costs via rapid random guessing, thereby enforcing the active exploration required to solve the task."
>
> - What is a “training attempt” (lines 262, 306)? Are the authors referring to a single run with a particular seed?
> A: This is correct. With "training attempt" we refer to an entire training run which is terminated when the training plateaued for 5 M episodes, encountered catastrophic forgetting or (hypothetically) successfully learned the task. As to the detail about the 'particular seed,' please refer to the question from above.
>
> - PPO and imitation learning methods are referred to as “one-shot” (lines 361-362). What do the authors mean by this? To me, they are anything but “one-shot”.
> A: We completely agree that this a poor choice of wording and is confusing, especially in the context of machine learning. We have removed the wording entirely.

---

### Author Response · Authors · 2025-12-02
**Thank you for your comments!**

We sincerely thank the reviewers for their engagement with our work. We highly appreciate the insightful questions regarding our baseline comparisons, reward structure, and the implications of our curriculum design. Addressing these comments has allowed us to refine our arguments and better contextualize our contributions within the broader reinforcement learning landscape. We will be addressing each raised question as a response to the author directly.Thank you!

---

### Meta-Review · Area_Chair_M4a5 · 2026-01-07

**Summary:**

This paper explores the value of human-driven curriculum learning for reinforcement learning, evaluated on a 3D Same-Different visuospatial task.

While the domain is interesting and seemingly novel, the presentation lacks formalism and the investigation lacks rigor, which calls into question the generality of the findings. Of note is the exclusion of experimental runs that "did not train due to bad random initialization". This is not good science, and the authors should not be cherry-picking seeds when reporting results.

It is worth metioning that most reviewers found the task quite interesting and novel for RL (reviewer exxm said "It’s refreshing to see a problem that feels genuine rather than over-engineered."). As such, the authors might want to consider reframing this as a new _challenge_ for the RL community, with their human-driven CL approach as a baseline.

**Reviewer Concerns:**

## exxm
- W1 (paper does not clearly formalize the task as a POMDP) was left unaddressed
- W2 (reward structure being -1/+1) was only verbally justified as to why they chose those values, which puts into question the generality of the findings. Additional ablation experiments could have been more informative.
- W3 (excluding experimental runs) is a serious concern and was not properly addressed.
- W4 (no clear experimental protocol) was only partially addressed, but not to complete satisfaction (in my opinion).
- W5 (justification for "expert demonstrations") was only partially addressed, but not to complete satisfaction (in my opinion).
- W8/W9 (encoder ablations) not properly addressed
- W10/W11 (shallow experimental comparison, considering only PPO, BC, and GAIL), not addressed.

## 1zTK
- "not obvious what we can extrapolate from this for RL in general": not addressed
- "it is likely that the apparent small gain in performance from the "human-inspired" curriculum was coincidence": not directly addressed
- "What is the exact action space?" While the authors did provide a clear answer here, it highlights the very specific design choices made, and calls into question the generality of their findings (similar to the issue with the choice of reward function).

## tQMc
- This reviewer provided a number of concrete concerns with regards to writing (relationship between paper and reward design, BC loss, etc.), which the authors did not respond to.
- Requested a more detailed discussion section, which the authors did not respond to
- Also raised issue of cherry-picking results, which the authors did not address.

## 2ZVc
- The reviewer provided three paragraphs under **Weaknesses** with a number of valid concerns (many already raised by other reviewers), which the authors did not respond to. The authors seem to have limited their responses to what was put under **Questions**.

**Reviewer Scores:**

- **exxm:** currently at 2, unlikely to increase
- **1zTK:** currently at 4, unlikely to increase
- ** tQMc:** currently at 2, unlikely to increase
- ** 2ZVc:** currently at 2, unlikely to increase

---

### Decision · Program_Chairs · 2026-01-26

Reject